# Acute Toxicity of the Dinoflagellate *Amphidinium carterae* on Early Life Stages of Zebrafish (*Danio rerio*)

**DOI:** 10.3390/toxics11040370

**Published:** 2023-04-13

**Authors:** Xiao Yang, Zhi Yan, Jingjing Chen, Derui Wang, Ke Li

**Affiliations:** 1Yantai Institute of Coastal Zone Research, Chinese Academy of Sciences, Yantai 264003, China; yangxiao20@mails.ucas.ac.cn (X.Y.); yanzhistudy@s.ytu.edu.cn (Z.Y.); 2022081549@stu.bzmc.edu.cn (J.C.); wangderui1999@163.com (D.W.); 2College of Resources and Environment, University of Chinese Academy of Sciences, Beijing 100049, China; 3School of Ocean, Yantai University, Yantai 264005, China; 4School of Pharmacy, Binzhou Medical University, Yantai 264003, China; 5College of Marine Science, Beibu Gulf University, Qinzhou 535011, China; 6Center for Ocean Mega-Science, Chinese Academy of Sciences, Qingdao 266071, China

**Keywords:** benthic harmful algal bloom, *Amphidinium carterae*, amphidinol, zebrafish, embryotoxicity

## Abstract

Dinoflagellates of the genus *Amphidinium* can produce a variety of polyketides, such as amphidinols (AMs), amphidinoketides, and amphidinin, that have hemolytic, cytotoxic, and fish mortality properties. AMs pose a significant threat to ecological function due to their membrane-disrupting and permeabilizing properties, as well as their hydrophobicity. Our research aims to investigate the disparate distribution of AMs between intracellular and extracellular environments, as well as the threat that AMs pose to aquatic organisms. As a result, AMs containing sulphate groups such as AM19 with lower bioactivity comprised the majority of *A. carterae* strain GY-H35, while AMs without sulphate groups such as AM18 with higher bioactivity displayed a higher proportion and hemolytic activity in the extracellular environment, suggesting that AMs may serve as allelochemicals. When the concentration of extracellular crude extracts of AMs reached 0.81 µg/mL in the solution, significant differences in zebrafish embryonic mortality and malformation were observed. Over 96 hpf, 0.25 μL/mL of AMs could cause significant pericardial edema, heart rate decrease, pectoral fin deformation, and spinal deformation in zebrafish larvae. Our findings emphasized the necessity of conducting systematic research on the differences between the intracellular and extracellular distribution of toxins to gain a more accurate understanding of their effects on humans and the environment.

## 1. Introduction

Benthic marine dinoflagellates, which are essential components in benthic food webs, play a variety of vital roles in coastal zones, especially in tropical and subtropical seas [1]. Dinoflagellates are generally notorious for their propensity to produce benthic harmful algal blooms (BHABs) containing marine biotoxins [2], which have a global impact on public health and aquatic ecological sustainability [3,4,5], thereby attracting the attention of the SCOR-IOC Global Ecology and Oceanography on Harmful Algal Blooms (GEOHAB) program [6]. Lipophilic polyether phycotoxins from benthic dinoflagellates, such as palytoxin [7], ciguatoxins [8], and okadaic acid [9], may act as allelochemicals to help dinoflagellates gain a competitive advantage over benthic organisms [10,11], posing serious threats to human health when accumulated in seafood through the marine food chain [12,13].

*Amphidinium carterae* is one of the most common species found in intertidal and aquaculture pond sediment in tropical, subtropical, and temperate ecosystems [14,15]. In its late growth stage, it generates numerous amphidinols (AMs) that can kill blood cells [16,17] and releases them into the environment [18]. AMs, which benefits from membrane disrupting and permeabilizing properties [19], are advantageous in studies of drug design, with dose-dependent bioactivities in potent anticancer and antifungal [20,21] and high toxicities in hemolytic activities [17,22], and have piqued the interest of natural product chemists worldwide [23]. Since the discovery of AM1 from *A. klebsii* in 1991 [24], a group of hairpin-shaped polyketides with similar core units [25], including more than 26 analogues, have been discovered [26]. However, the hydrophobicity of the polyene has a significant impact on the bioactivity of AMs [20,27]. AM25’s highly hydroxylated branches and substituent disulfate ester group increase its polarity and significantly reduce its hemolytic activities against seabream erythrocytes [26] in comparison to AM3, its most potent homologue (Figure 1) [28]. Therefore, variations of AMs in the distribution patterns between intracellular and extracellular may result in distinct ecological effects.

Using the liquid chromatography-tandem mass spectrometry (LC-MS/MS) method, the difference in AM proportions between intracellular and extracellular environments of *A. carterae* were compared [29,30]. By comparing the hemolytic activity of AMs intracellularly and extracellularly at the same mass concentration, we determined the influence of different distribution patterns on the hemolytic activity. Recent research indicates that waterborne toxins in seawater impair the sensorimotor function of fish larvae [31,32]. Recent investigations have strengthened the notion that zebrafish embryos represent a valuable model to study the ecotoxicological and developmental toxicity of environmental pollutants and have been recognized by the OECD. The mature zebrafish embryo model provides a new insight to evaluate microalgal toxicity in ocean and freshwater and the changes in toxicity induced by different environmental conditions using acute toxicity tests [33,34]. On the other hand, AMs from *Amphidinium* have good antitumor activity and medicinal value. Zebrafish embryo acute toxicity tests as a pharmacological model of marine natural products to predict mammalian developmental toxicity are gaining acceptance [35]. Therefore, we assessed the acute toxicity of extracellular AMs on zebrafish embryos to reflect the potential threat of *A. carterae* to aquatic organisms, thus increasing our understanding of the substantial harm posed by this type of polyether toxin to the development of the fish species in the natural environment.

## 2. Materials and Methods

### 2.1. Culture of A. carterae

An axenic strain of *Amphidinium carterae* Hulburt (GY-H35, which originated from the China East Sea) was purchased from the Shanghai Guangyu Biological Technology Co., Ltd. (Shanghai, China) and maintained in f/2 medium at 20 °C and 4000 LUX in a 12:12 h LD cycle [17]. In a two-liter autoclaved conical flask, *A. carterae* cells in the exponential growth phase were inoculated with 1 liter of filter-sterilized f/2 medium at an initial concentration of 5000 cells mL^−1^. The density of cells was determined using a blood count plate and counting more than 100 cells under an inverted microscope (Olympus CKX53, Tokyo, Japan). During the stationary phase, the biomass was harvested (the final concentration was 2.4 × 10^5^ cells mL^−1^) by centrifugation for 10 min at 4 °C at 2300 rpm [21]. The supernatants and cell pellets were stored at −80 °C separately until chemical extraction.

### 2.2. Preparation of A. carterae Toxin

To obtain the intracellular toxin, the cell pellets were resuspended in 20 mL of cold methanol in an amount up to twice the biomass, and then cryogenic sonication treatment was applied twice for 10 min. The sonication was repeated until complete cell disruption was confirmed using an inverted microscope (Olympus CKX53, Tokyo, Japan). Next, 20 mL CH_2_Cl_2_ and 10 mL distilled water were added to the crude MeOH extract, and the mixture was ultrasonically shocked for 30 min and left to stand for 12 h at 4 °C [36]. After the solution was completely stratified, the upper solution was taken and was centrifuged at 6700× *g* for 5 min, and the supernatant was filtered through a 0.2 µm syringe filter (syringe-driven filter; JET BIOFIL, Guangzhou, China), dried in vacuum, and stored at −80 °C.

To acquire the extracellular toxin, the culture’s supernatant was enriched by passing it through C_18_ solid phase extraction (SPE) cartridges (200 mg, Waters, Milford, MA, USA) [37]. Briefly, the cartridges were conditioned with 2 mL of methanol and then equilibrated with 2 mL of 50% aqueous methanol and 2 mL of deionized water. Samples were loaded into equilibrated SPE cartridges, which were then rinsed with 2 mL of deionized water and 50% aqueous methanol. The extracellular toxin was then eluted with 100% methanol, dried in vacuum, and stored at −80 °C.

From 1 liter of culture, intracellular toxin (91.2 mg) and extracellular toxin (16.2 mg) were extracted and dissolved in 2 mL of MeOH for biological activity assays.

### 2.3. Identification of AMs

#### 2.3.1. Conditions for Chromatographic Separation

The high-resolution ultra-high-performance liquid chromatography and high-resolution mass spectrometry (UPLC-HR-MS) system consisted of an Acquity UPLC system and a time-of-flight mass spectrometer along with an electrospray ionization source (Waters, Milford, MA, USA). UPLC separation was achieved on a BEH C_18_ column (100 mm × 2.1 mm, 1.7 μm particle size) (Waters, USA). The mobile phase flow rate was 0.25 mL/min, with the column thermostatically set at 18 °C. The mobile phase consisted of water with FA, 0.1% (A) and methanol (B). The gradient elution was as follows: 0–1 min, 95% A; 1–4 min, 40% A; 5–6 min, 40% A; and 6–10 min, 95% A. The volume of the injected sample was 10 μL.

#### 2.3.2. Conditions for High-Resolution Mass Determination

Negative electrospray ionization was used for HRMS. Spectra in the *m*/*z* 100–2000 range were captured for the full-scan MS analysis. The collision gas was argon at a pressure of 5.3 × 10^5^ Torr, while nitrogen was utilized as the desolvation gas (600 L/h) and cone gas (50 L/h). The capillary voltage and cone voltage were set to 30 V and 2.8 kV, respectively, while the source and desolvation temperatures were 100 and 400 °C, respectively. For the MS spectrum and the MS/MS spectrum, the collision energies for collision-induced dissociation were 10 and 80 eV, respectively. With an interscan delay of 0.5 s, the scan time was set to 0.2 s. Leucine enkephalin, at a concentration of 100 ng/mL, served as the dual electrospray ion source with internal references for the LockSprayTM system employed in these tests. The flow rate was set to 5 μL/min, and lock-mass calibration data at *m*/*z* 554.2615 in negative ion mode were obtained for 1 s every 10 s. The voltage setting was switched in a quasi-simultaneous fashion to produce nonselective collision-induced dissociation [38]. Five different voltages (5, 15, 25, 35, and 45 V) were applied, while distinct but parallel data collection processes were used for each potential (multiplexed nonselective CID).

### 2.4. Measurement of Hemolytic Activity

Rabbit erythrocytes were obtained from Shanghai Yuanye Bio-Technology Co., Ltd. (Shanghai, China). Erythrocytes were washed, centrifuged, and diluted with Alsever’s solution to a final concentration of 1% (*v*/*v*). Each assay was carried out in triplicate in a 1.5 mL centrifuge tube by mixing 0.5 mL of blood with 0.5 mL of diluted AMs after incubation for 2 h at 35 °C in the dark, and the centrifuge tubes were centrifuged at 1200× *g* for 3 min at 4 °C. Hemolysis controls for 0 and 100% lysis were prepared by mixing 0.5 mL of blood with 0.5 mL of Alsever’s solution or distilled water, respectively. From the tube, 200 µL of the supernatant was transferred to a 96-well plate and the absorbance at 414 nm was measured with a microplate reader (Multiscan GO, Thermo Fisher Scientific Inc., Waltham, MA, USA).

### 2.5. Zebrafish Maintenance and Embryo Collection

The Yantai Institute of Coastal Zone Research, Chinese Academy of Sciences, Animal Care Ethics Committee (2021R001) guidelines were followed for all animal experiments. Adult wild AB zebrafish (*Danio rerio*) were obtained from the Department of Shanghai FishBio Co., Ltd. (Shanghai, China) and kept in a flow-through aquarium system with a 14:10 h light:dark photoperiod at 28 ± 0.5 °C, as described in The Zebrafish Book [39]. The fish were fed with *Artemia* nauplii twice per day. The medium of culture was completely replaced every 24 h. Healthy female and male fish were placed on either side of the mating box and isolated overnight. The obstacle was removed at 8 a.m. the following morning, with unfertilized and dead embryos being picked out and examined under an inverted microscope (Olympus CKX53, Japan) [40].

### 2.6. Zebrafish Embryo Acute Toxicity Tests

According to the pre-experiment data, extracellular toxin concentrations (0.081, 0.81, 1.62, 2.03, and 2.43 µg/mL) were tested in five concentrations. The wild-type zebrafish embryos (4 h post-fertilization, 4 hpf) were examined using an inverted microscope (Olympus CKX53, Japan). Healthy embryos were then distributed in 12-well plates for 96-hpf exposure experiments, with 10 embryos in approximately 5 mL exposure solution per well in 3 exposure replicates, as well as a solvent control treatment (0.03% MeOH) and a negative control treatment (sea salt solution with a conductivity of 500~550 µs/cm). The exposure solution was completely replaced every 24 h. The bath temperatures in the test chambers were kept constant throughout the tests at 28 ± 0.5 °C.

During the trial, the number of dead, deformed, and hatching embryos in individual concentrations was recorded according to OECD Test Guideline (TG) 236. The coagulation of fertilized eggs, the absence of somite development, the non-detachment of the tail, and the absence of a heartbeat are all thought to be signs of embryonic death [41]. According to the mortality results, the median lethal concentration (LC_50_) was calculated using Probit analysis and SPSS 10.0 statistical software (SPSS, Chicago, IL, USA). The embryo’s spontaneous movements at 24 hpf, the heart rate of the embryos or hatched larvae at 48, 72, and 96 hpf, and the body length of the hatched larvae at 96 hpf were tested. To investigate cardiotoxicity, the size of pericardium edema (PE) and heartbeat were measured using the previously published method [42,43].

### 2.7. Statistical Analysis

Data are expressed as the mean ± SD of at least three independent experiments. The statistical significance of differences was evaluated by standard one-way ANOVA and chi-square test (SPSS Inc., Chicago, IL, USA). An asterisk or different letters denoted statistical significance for values of *p* < 0.05, while two asterisks denoted highly significant values of *p* < 0.01. NS was regarded as having no significant difference.

## 3. Results and Discussion

### 3.1. Distribution Pattern of AMs between Extracellular and Intracellular

To investigate the possible AMs, intracellular toxin and extracellular toxin were detected using electrospray ionization (ESI) mass spectrometry in negative ion mode. The result of a single mass composition study (Appendix A) and collision-induced dissociation with high resolution mass spectrometric (HRMS) analysis (Table 1) led to the discovery of three known AMs and three novel AM derivatives. The AM18 was deduced from the observed adduct [M–H]^−^ (*m*/*z* 1357.8248; calculated *m*/*z* 1357.8248, Δ = 0.0 ppm), with a molecular formula of C_71_H_122_O_24_ (degrees of unsaturation = 11). The AM19 sulfated from AM18 was inferred from the observed adduct [M−H]^−^ (*m*/*z*, 1437.7759; calculated *m*/*z*, 1437.7816, Δ = −4.0 ppm), with a molecular formula of C_71_H_122_O_27_S (degrees of unsaturation = 11). The AM2 was deduced from the observed adduct [M−H]^−^ (*m*/*z* 1373.8121; calculated *m*/*z* 1373.8197, Δ = −5.5 ppm), with a molecular formula of C_71_H_122_O_25_ (degrees of unsaturation = 11).

According to their chemical structures, AMs from *Amphidinium* species had molecular weights of more than 1000 Da and included the same core unit, two tetrahydropyran rings separated by a C6 chain, a polyunsaturated (lipophilic) alkyl arm, and a long polyhydroxy (hydrophilic) arm [25]. Consequently, identical chemical structures resulted in comparable fracture patterns. The AM sodium adducts formed a cleavage at the C–C bond in the α position of the tetrahydropyran ring B [29]. In negative ion mode, we found fragments with *m*/*z* 661.4, *m*/*z* 577.3 (AM18, Figure 2A), *m*/*z* 1019.5, *m*/*z* 1031.5 (AM19, Figure 2B), and *m*/*z* 577.3 (AM2, Figure 2C) that share the same cleavage by losing the lipophilic arm. Furthermore, the ε position of the tetrahydropyran ring B was also a unique cleavage site, producing the fragmentations of *m*/*z* 241.2 (AM18, Figure 2A) and *m*/*z* 1139.7 (AM2, Figure 2C). Meanwhile, the cleavage of the β positions of the carbonyl group led to the loss of the lipophilic arm, resulting in the fragmentations of *m*/*z* 1139.7, *m*/*z* 1081.6, *m*/*z* 275.2 (AM18, Figure 2A), *m*/*z* 297.1, and *m*/*z* 355.1 (AM19, Figure 2B). The molecular ions of dehydrogenated aldehyde, such as *m*/*z* 297.1, *m*/*z* 483.2, *m*/*z* 527.3, and *m*/*z* 629.3, evidenced that AM19 contained the sulfate ester group structure. Three unidentified AM derivatives (AD1, AD2, and AD3) were equivalent to AM1, AM27, and AM11, respectively. Although their fragmentation pattern and HRMS analysis did not match any known AMs, the featured *m*/*z* 661.1 indicated that they all belonged to the AM family, and the presence of *m*/*z* 297.1 in AD1 and AD3 indicated the existence of sulfate ions.

Although most studies focused on AMs, their distribution between intracellular and extracellular environments had rarely been explored, which is very important to understanding their harmful effects and ecological function. Under the HPLC condition, the six AMs mentioned above were well separated, and the percentages were evaluated from the HPLC peak areas. As shown in Table 2, there were significant differences in the proportion of different AMs in the extracellular and intracellular environments. AM19 occupied the highest proportion of the total of the six AMs in either intracellular (99.99%) or extracellular (99.72%) extracts. AD1 had the highest proportion in the extracellular AM fraction (0.23%) and the lowest proportion in the intracellular AM fraction (7.91 × 10^−4^%). Furthermore, we found that higher bioactive sulfur-free AMs had a higher proportion in the extracellular AM fraction (3.86 × 10^−2^%), which indicated that the extracellular AMs might have stronger bioactivity at the same mass concentration. The hemolytic activity experiment with rabbit erythrocytes proved our inference, and the EC_50_ (7.33 µg/mL, 7.08–7.55 µg/mL) of extracellular crude extracts of AMs was higher than that of intracellular extracts (12.09 µg/mL, 11.53–12.67 µg/mL).

### 3.2. Mortality and Hatching Rates of Zebrafish Embryos Treated with A. carterae Extracellular AMs

In our experiment, the difference between the control and the solvent control for all indicators was non-significant, and the 100% hatching rate met the requirements of OECD No. 236 [37], which indicated that the highest MeOH concentration of 0.03% had no toxicity or weak toxicity to the development of zebrafish embryos. Therefore, the experiment data from the solvent were used as a control.

Embryotoxicity was assessed at five different concentrations of extracellular AMs (0.081, 0.81, 1.62, 2.03, and 2.43 µg/mL lower than culture concentration at 16.2 µg/mL). As shown in Table 1, zebrafish embryos began to exhibit significant lethal toxicity when the concentration of toxin exceeded 0.81 µg/mL, with up to 73.33% mortality at higher concentrations (2.43 µg/mL). The LC_50_ at 12, 48, 72, and 96 hpf was analyzed using SPSS software, as shown in Table 3. Among them, the LC_50_ at 12 hpf, which was 2.07 µg/mL, was the largest, while the LC_50_ at 96 hpf, which was only 1.49 µg/mL, was the smallest. There was an obvious dose–effect relationship between zebrafish embryo mortality and concentration at different time periods (*p* < 0.01 by chi-square test), indicating that the toxicity of AMs appears to be concentration- and time-dependent in zebrafish embryos. As shown in Figure 3, there was no significant difference in hatching rate between the experimental groups at 72 hpf.

### 3.3. Teratogenicity of Zebrafish Embryos Treated by A. carterae Extracellular AMs

Based on an acute toxicity experiment, AMs had sublethal effects on early-life stages of zebrafish and induced a series of deformities during development by different dosages, including abnormal spontaneous movement, slow heart rate, hypopigmentation, and morphological deformities. In acute zebrafish embryo toxicity, hemolytic toxic compounds produced by two other harmful dinoflagellates, *Karenia mikimotoi* [44] and *Karlodinium veneficum* [45], exhibited similar teratogenic effects, such as edema and spinal and tail deformation. It is worth noting that zebrafish larvae exposed to low-concentration karlotoxins produced by *K. veneficum* showed evident epithelium damage through apoptosis, and microcystin-LR also decreased heart rate, though it triggered apoptosis in the heart of zebrafish embryos [46]. In our experiments, the AMs might also induce the pectoral fin deformities and decreased heart rates of zebrafish larvae by inducing apoptosis. However, the specific cause of apoptosis remains to be further studied. As shown in Figure 4, multiple morphological deformities in zebrafish at 72 hpf were observed, and the rate of deformity is shown in Figure 5A. The EC_50_ for 72 hpf is 1.04 µg/mL (0.83–1.22 µg/mL).

The spontaneous movement of zebrafish is primarily the wobble of the tail, which is innervated by the primary neurons and independent of the brain neurons of the zebrafish. This is the earliest detectable neural behavior, an important indicator for evaluating underlying neuromuscular abnormalities [47]. At 24 hpf, the frequency of spontaneous movement of embryos exposed to high concentrations of AMs (1.62, 2.03, and 2.43 µg/mL) was significantly lower than that of the control group (Figure 5B), indicating that AMs were neurotoxic to zebrafish embryos at an early stage. In addition, body lengths decreased to 3.30 mm when exposed to the 2.43 µg/mL AMs at 96 hpf, compared to 3.74 mm in the control group (Figure 5C).

When the exposure concentration of AMs reached 1.82 µg/mL, obvious cardiac malformations were observed. The morphological changes of the heart were pericardial edemata and a lack of heart tube looping (Figure 5F), and the impaired cardiac function was a decrease in heart rate (Figure 5E). To better examine the heart malformation in embryonic zebrafish exposed to AMs, the areas of pericardium were measured to provide an index of the edema size. At 96 hpf, the areas of pericardium were 0.015 mm^2^ in the control, which were significantly increased to 0.019 and 0.025 mm^2^ in the 0.81 and 2.03 µg/mL, respectively (Figure 5E). These findings demonstrated that AMs induce dose-dependent embryonic developmental toxicity.

If it is considered that, in an *A. carterae* bloom, cell densities can reach 2 × 10^5^ cells/mL [48] and approximately 16.2 mg/L of secretions containing AMs are in the medium, that is significantly higher than the lethal concentration for a zebrafish embryo at 96hpf. To date, only one AM derivative, carteraol E, isolated from lab-cultured *A. carterae,* has been proven to be a polyhydroxyl ichthyotoxin [49], which may play a similar ecological role as karlotoxins produced by species of *Karlodinium* structurally similar to AMs [45,48]. However, carteraol E was not detected in the GY-H35 strain of *A. carterae* utilized in this study. Thus, our findings validated the lethal and sublethal effects of AMs with hemolytic activity from *A. carterae* on embryonic development in zebrafish.

We have shown that *A. carterae* produces AMs differently in intracellular and extracellular environments and that this distribution difference is related to the various biological roles. To better understand the potential environmental concerns during blooms, all the chemicals that have an allelopathic function need to be identified because different AM concentrations with various levels of activity may have adverse impacts on various aquatic creatures in different ways. In addition, it is still unknown how AMs are produced and secreted, whether sulfation protects cells from toxicity, and how environmental variables such as temperature and nutrition affect AM synthesis and sulfation. Furthermore, the lethal and teratogenic toxicity of AMs to zebrafish embryos highlighted the need to simultaneously assess the biological toxicity and anticancer efficacy of AMs as well as their potential medical utility.

## 4. Conclusions

The distribution of harmful dinoflagellates in tropical and subtropical seas will expand as a result of global warming [50], which will have a negative impact on aquaculture and environmental safety. Our study highlights differences in the proportion and distribution of biotoxins released by harmful bloom algal species between intracellular and extracellular environments in order to better monitor and comprehend their ecological significance. In this experiment, we found the allelochemical roles of AMs, which, with higher hemolytic activity, occupied a higher proportion in the extracellular environment, while less active sulfated AMs were more intracellular to protect the algae from toxicity. AMs secreted extracellularly exhibited significant lethal and sublethal toxicity to the embryonic development of zebrafish. When the concentration of extracellular AMs reached 0.81 µg/mL (lower than the concentration in the culture medium), embryonic mortality was significantly increased and showed deformities such as pericardial edema and spinal deformation.

## Figures and Tables

**Figure 1 toxics-11-00370-f001:**
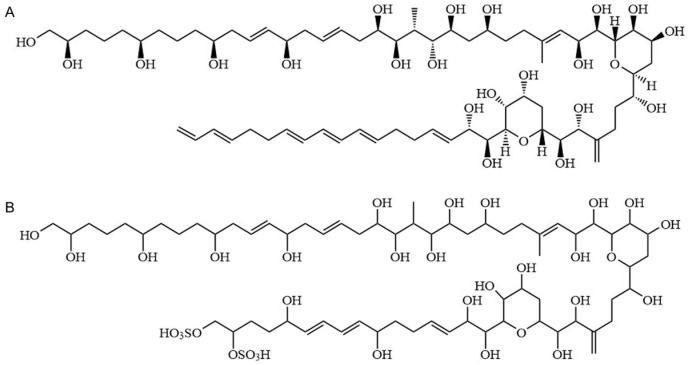
Chemical structures of amphidinol 3 (AM3, **A**) and amphidinol 25 (AM25, **B**).

**Figure 2 toxics-11-00370-f002:**
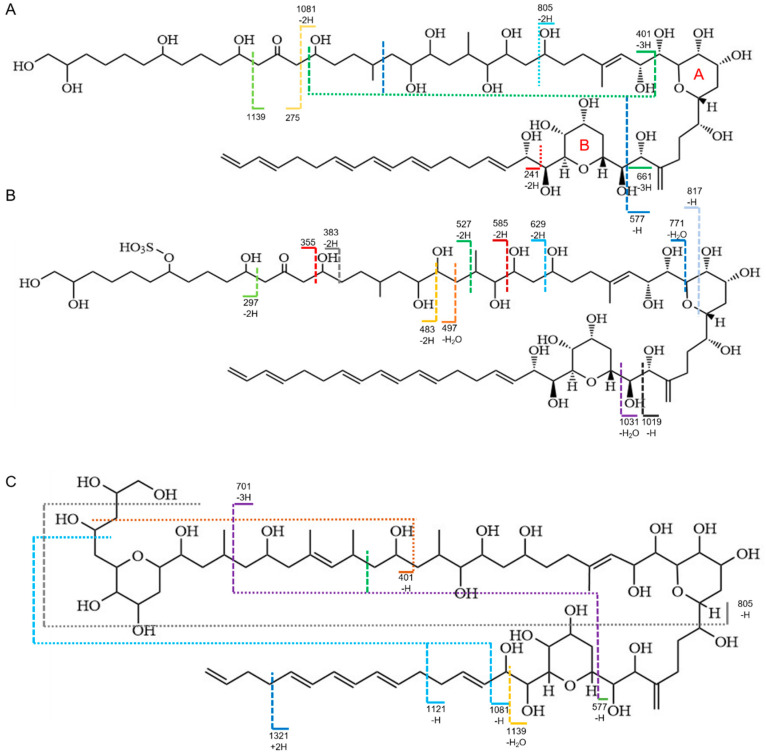
Main MS/MS fragments observed for amphidinol 18 (**A**), amphidinol 19 (**B**), and amphidinol 2 (**C**).

**Figure 3 toxics-11-00370-f003:**
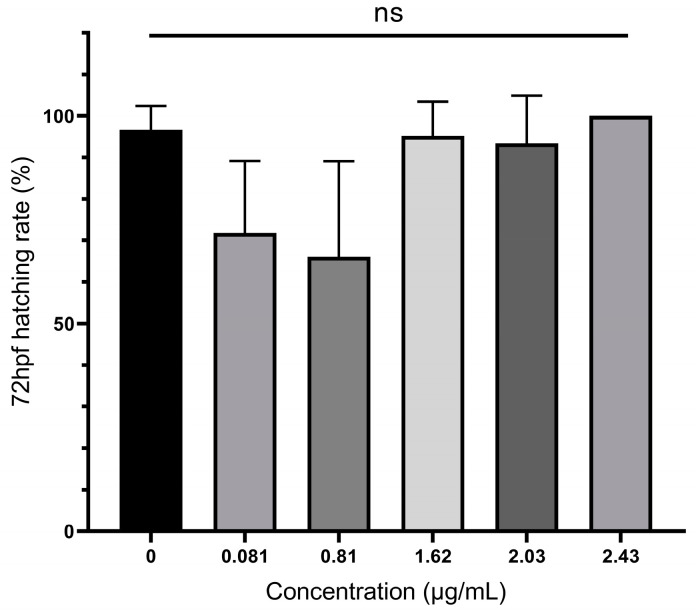
The hatching rate of zebrafish embryos exposed to AMs for each observation time. Hatching rate was calculated with the equation (number of hatched individuals)/(survival individuals of one replicate) and represented as the mean ± SD (n = 3). Tukey’s multiple comparison test and one-way ANOVA were used to examine the results of the experiments. NS was regarded as having no significant difference.

**Figure 4 toxics-11-00370-f004:**
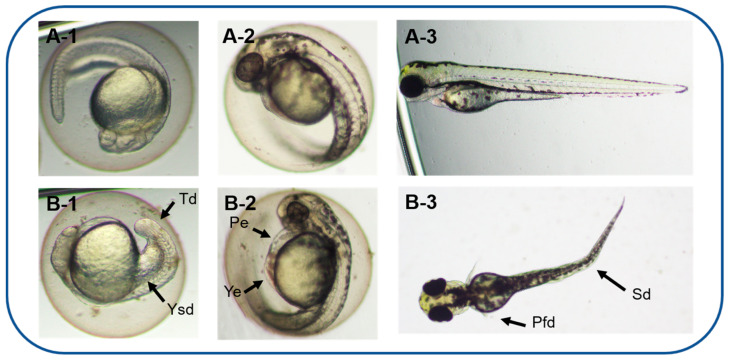
Morphological abnormalities induced by AMs from *A. carterae* at different times. The normal zebrafish embryos in the control group were at 24 hpf (**A-1**), 48 hpf (**A-2**), and 72 hpf (**A-3**). Typical malformations caused by 2.03 µg/mL of AMs on zebrafish embryonic development at 24 hpf (**B-1**), 48 hpf (**B-2**), and 72 hpf (**B-3**). Abbreviations: Pe, pericardial edema; Pfd, pectoral fin deformities; Sd, spinal deformation; Td, tail extension deformity; Ye, yolk edema; Ysd, yolk sac deformity.

**Figure 5 toxics-11-00370-f005:**
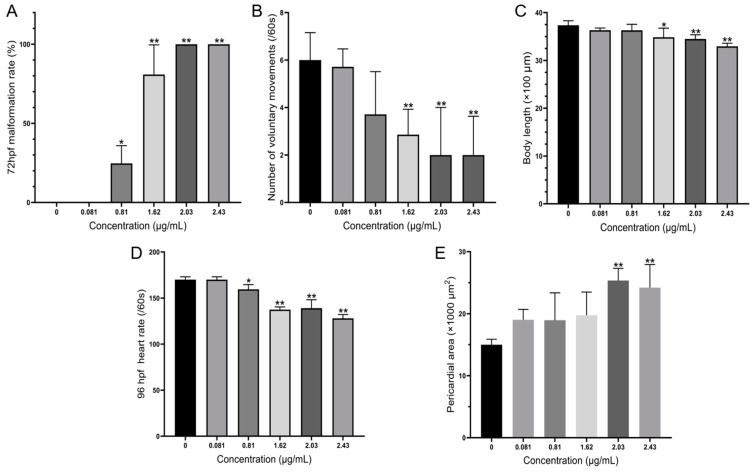
(**A**) Malformation rate at 72 hpf; (**B**) number of spontaneous movements per minute at 24 hpf; (**C**) body length of hatched individual at 96 hpf; (**D**) heart rate per minute at 96 hpf; (**E**) pericardium area at 96 hpf. Values represented averages from four replicates with standard deviation as error bars. An asterisk indicates a significant difference between the treatment and the control (one-way ANOVA, Turkey’s HSD at *p* < 0.05 denoted by *, *p* < 0.01 denoted by **).

**Table 1 toxics-11-00370-t001:** Exact masses, elemental composition, and error between empirical and theoretical masses of AMs.

ID		*m*/*z*	Formula	Calc.	mDa	ppm
AM18	1	1357.8248	C_71_H_122_O_24_	1357.8248	0.0	0.0
	2	1139.6733	C_60_H_100_O_20_	1139.6730	0.3	0.3
	3	1081.6312	C_57_H_94_O_19_	1081.6311	0.1	0.1
	4	805.4382	C_43_H_66_O_14_	805.4374	0.8	1.0
	5	661.3799	C_33_H_58_O_13_	661.3799	0.2	0.3
	6	577.3227	C_28_H_50_O_12_	577.3224	0.3	0.5
	7	561.3064	C_31_H_46_O_9_	561.3064	0.0	0.0
	8	401.2540	C_21_H_38_O_7_	401.2539	0.1	0.2
	9	275.1856	C_14_H_28_O_5_	275.1858	−0.2	−0.7
	10	241.1533	C_17_H_22_O	241.1440	−5.9	−24.5
AM19	1	1437.7759	C_71_H_122_O_27_S	1437.7816	−5.7	−4.0
	2	1031.5466	C_48_H_88_O_21_S	1031.5461	0.5	0.5
	3	1019.5461	C_47_H_88_O_21_S	1019.5461	0.3	0.3
	4	771.4200	C_36_H_68_O_15_S	771.4201	−0.1	−0.1
	5	629.3210	C_28_H_54_O_13_S	629.3207	0.3	0.5
	6	585.2947	C_26_H_50_O_12_S	585.2945	0.2	0.3
	7	527.2529	C_23_H_44_O_11_S	527.2526	0.3	0.6
	8	497.2423	C_22_H_42_O_10_S	497.2420	0.3	0.6
	9	483.2266	C_21_H_40_O_10_S	483.2264	0.2	0.4
	10	383.1382	C_15_H_28_O_9_S	383.1376	0.6	1.6
	11	355.1435	C_14_H_28_O_8_S	355.1427	0.8	2.3
	12	297.1017	C_11_H_22_O_7_S	297.1008	0.9	3.0
AM2	1	1373.8121	C_71_H_122_O_25_	1373.8197	−7.6	−5.5
	2	1321.8035	C_67_H_118_O_25_	1321.7884	16.0	12.1
	2	1139.6661	C_56_H_100_O_23_	1139.6577	8.4	7.4
	3	1121.6622	C_56_H_98_O_22_	1121.6471	15.1	13.5
	4	1081.6318	C_53_H_94_O_22_	1081.6158	16.0	14.8
	5	701.4111	C_36_H_62_O_13_	701.4112	−0.1	−0.1
	6	577.3232	C_28_H_50_O_12_	577.3224	0.8	1.4
	7	401.2544	C_21_H_38_O_7_	401.2539	0.5	1.2
AD1	1	1487.7990	C_73_H_125_O_27_SNa	1487.7948	4.2	2.8
	2	701.4068	C_36_H_62_O_13_	701.4112	−4.4	−6.3
	3	661.3774	C_33_H_58_O_13_	661.3799	−2.5	−3.8
	4	577.3196	C_28_H_50_O_12_	577.3224	−7.8	−13.5
	5	355.1416	C_14_H_28_O_8_S	355.1427	−1.1	−3.1
	6	297.1006	C_11_H_22_O_7_S	297.1008	−0.2	−0.7
AD2	1	1183.6519	C_54_H_100_O_25_	1183.6475	4.4	3.7
	2	661.3774	C_33_H_58_O_13_	661.3799	−2.5	−3.8
AD3	1	1453.7662	C_71_H_122_O_28_S	1453.7765	−10.3	−7.1
	2	701.4092	C_36_H_62_O_13_	701.4112	−2.0	−2.9
	3	661.3794	C_33_H_58_O_13_	661.3799	−0.5	−0.8
	4	577.3221	C_28_H_50_O_12_	577.3224	−0.3	−0.5
	5	355.1422	C_14_H_28_O_8_S	355.1427	−0.5	−1.4
	6	275.1858	C_14_H_28_O_5_	275.1858	−7.9	−28.7
	7	297.1014	C_11_H_22_O_7_S	297.1008	0.6	2.0

**Table 2 toxics-11-00370-t002:** Proportion of different AMs between intracellular and extracellular environment.

AM	AM18	AM19	AM2	AD1	AD2	AD3
Extracellular	0.0061	99.7250	0.0248	0.2342	0.0078	0.0021
Intracellular	0.0009	99.9899	0.0048	0.0008	0.0023	0.0013

**Table 3 toxics-11-00370-t003:** Lethal effects of AMs from *A. carterae* on the early development of zebrafish embryos. The zebrafish embryos were treated with 0.03% MeOH (control), and 0.081, 0.81, 1.62, 2.03, and 2.43 µg/mL of AMs at different times of exposure (12, 24, 48, 72, and 96 h). Descriptive data represent the number of dead embryos as a percentage of three independent experiments (n = 30 per group, one-way ANOVA, Turkey’s HSD at *p* < 0.05 denotes *, *p* < 0.01 denotes **).

Time of Exposure (hpf)	Concentration (µg/mL)						LC_50_ (95% Confidence)	X^2^ (*p*)
	0	0.081	0.81	1.62	2.03	2.43		
12	0, 0, 0	0, 0, 1	2, 0, 4	5, 4, 2 *	5, 3, 4 *	6, 9, 7 **	2.07 (1.30–5.58)	49.97 (*p* < 0.001)
24	0, 0, 0	0, 0, 1	2, 4, 5 *	5, 4, 2 *	5, 3, 4 *	6, 9, 7 **	1.85 (1.27–3.15)	43.384 (*p* < 0.001)
48	0, 0, 0	0, 0, 1	2, 4, 5 *	6, 5, 3 **	5, 4, 5 **	6, 9, 7 **	1.539 (1.08–2.35)	48.57 (*p* < 0.001)
72	0, 0, 0	0, 0, 1	3, 4, 5 **	6, 5, 3 **	5, 4, 5 **	6, 9, 7 **	1.49 (1.04–2.28)	47.397 (*p* < 0.001)
96	0, 0, 0	0, 0, 1	3, 4, 5 *	6, 5, 3 **	5, 4, 5 **	6, 9, 7 **	1.49 (1.04–2.28)	47.397 (*p* < 0.001)

## Data Availability

Data are contained within the article and Appendix A.

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
