# Peer review of "Acute Toxicity of the Dinoflagellate Amphidinium carterae on Early Life Stages of Zebrafish (Danio rerio)"

_toxics, 2023, doi:10.3390/toxics11040370_

Round 1

Reviewer 1 Report

The work is relevant and used a good experimental design, however some points deserve clarification and additions.

1) It is important to mention in the introduction the advantages for the authors to have used Zebrafish and not another experimental animal model. I need to say more than has been said. What are the advantages.

2) the experimental design of separation by chromatography was standardized by the group or was based on some reference, important to mention in the text. Same for chromatography activity

3) Regarding the Toxicity Test. Although the annotations of mortality and anomalies were based on the OECD 236 test, the way in which the test was performed differed in some points from the OECD 236 test recommendations. hpf, embryos up to 3hpf, 5 embryos per well and up to 2ml in volume should be used.

Explain the modifications.

4) One of the advantages of using Zebrafish embryos and larvae is their transparency, which makes it possible to take photographs that portray the observed effects. I suggest new figures with photos of the abnormalities detected at different analysis times.

5) it is important to mention in the description of the results/discussion if there is any relationship between the results found for hemolysis and damage to Zebrafish embryos/larvae.

6) explain why some analyzes/results were placed at 72hpf and others at 96hpf

7) it is necessary to discuss the results obtained with Zebrafish, the anomalies detected in view of what already exists in the literature
